# Patient safety culture in public hospitals of Ethiopia: A systematic review and meta-analysis

Gedion Asnake Azeze[1], Kirubel Eshetu Haile[2]*, Amanuel Yosef Gebrekidan[3], Gizachew Ambaw Kassie[3], Berhan Tsegaye Negash[1], Yordanos Sisay Asgedom[3]

1 Department of Midwifery, College of Medicine and Health Science, Hawassa University, Sidama region, Ethiopia, 2 School of Nursing, College of Health Science and Medicine, Wolaita Sodo University, Wolaita Sodo, Ethiopia, 3 School of Public Health, College of Health Science and Medicine, Wolaita Sodo University, Wolaita Sodo, Ethiopia

* hailekirubeleshetu@gmail.com

## Abstract

### Background

Patient safety is a major health care challenge and is an issue of global concern. Poor healthcare system in Ethiopia, alongside increasing concerns for safety and patient safety culture, prompted to this systematic review and meta-analysis. Our systematic review and meta-analysis aims to determine the pooled prevalence of patient safety culture among healthcare providers in Ethiopia from studies that administered Hospital Survey on Patient Safety Culture (HSOPSC) tool.

### Method

This study employed a systematic review and meta-analysis and followed the Preferred Reporting Items for Systematic reviews and Meta-Analysis (PRISMA) guidelines. We carried out a comprehensive search for studies in Science Direct, Medline, African Journals Online (AJOL), Excerpta Medical Database (EMBASE), Scopus and Google Scholar to identify relevant observational studies published in the English language. Data sources were searched on the 10th of October 2024 and updated the search on the 29th of March 2025. All selected articles met the inclusion criteria according to the Participants, Intervention, Comparison and Outcome (PICOS) criteria.. Random effects model meta-analysis using STATA version 15 statistical software was conducted to estimate the pooled prevalence with 95% confidence intervals (CI). Funnel plot and $I^2$ test were used to check publication bias and heterogeneity between studies, respectively. Subgroup analysis and leave-out-one sensitivity analysis was done.

### Results

Searches retrieved a total of 667. After removing duplicates, 610 remained. 138 studies were reviewed in full, but only 14, which included 5,901 health care providers,

**Data availability statement:** All relevant data are within the paper and its Supporting Information files.

**Funding:** The author(s) received no specific funding for this work.

**Competing interests:** The authors have declared that no competing interests exist.

**Abbreviations:** HSOPSC, Hospital Survey on Patient Safety Culture; LMICs, low- and middle-income countries; NOS, Newcastle-Ottawa Scale; PSC, Patient Safety Culture

were considered in the final meta-analysis. The pooled prevalence of patient safety culture among health care providers in public hospitals of Ethiopia was 45.52% (95% CI: 41.27, 49.76; $I^2 = 91.3$; p-value<0.001).

## Conclusion and Implications

Patient safety culture among health care providers in Ethiopia remained poor. These findings underscore the need for increased focus on education and training, development of guidelines and policies for patient safety to integrate a patient safety culture into the existing health system. Additionally, integrating patient safety education and training topics early in pre-service curricula is crucial.

## Registration

Registered in PROSPERO database with registration number of CRD42023407601

## Background

Patient safety is defined as the reduction of risk of unnecessary harm associated with heath care to an acceptably minimum degree. Patient safety culture, on the other hand, is the product of individual and group values, perceptions, attitude, competencies and patterns of behavior that staffs share within an organization related to patient safety [1–3]. It is an extremely important factor and a crucial element in the effort to reduce disabling injuries and/or deaths that are directly related to medical care within hospital settings. The issue of patient safety was recognized as a significant topic in the global agenda during the 55th World Health Assembly [2,4–6]. Key medical practices and associated risks that compromise patient safety include hospital-acquired infections, diagnostic errors, unsafe injection practices, unsafe surgical care procedures, and medication errors, unsafe care in mental health, unsafe transfusion practices, radiation error and venous thromboembolism [7].

The identification and mitigation of medical errors has rapidly become an important concern within primary health care, representing a major priority for all healthcare practitioners [8,9]. Globally, it is estimated that nearly 40% of patients experiance harm in primary and outpatient healthcare settings, with as much as 80% of these incidents being preventable [7,10,11]. Evidence highlights that a minimum of one in ten patients in high-income countries experience safety failures or adverse events while receiving hospital care, with at least 50% of these incidents being preventable [7]. The estimate for low- and middle-income countries (LMICs) suggests that up to one in four patients are harmed [12,13]. Every year, hospitals in LMICs experience 134 million adverse events attributable to inadequate safety measures, leading to approximately 2.6 million deaths [11,14]. The situation is more challenging and serious in developing countries, where the risk of patient harm is greater due to inadequate infrastructure and limited resources [3].

Hospital staff in low-income settings identified challenges in maintaining patient safety. There primary concerns were associated with material context, staffing

challenges and inter-professional working relationships [15]. Evidences also shows several factors can compromise patient safety, including failed organizational processes, poor collaboration among team members, understaffing, extended working hours, shortage of basic medical equipment, lack of strong leadership in patient safety and the mental and physical strain experienced by healthcare professionals [2,16].

One of the ultimate goals of the Ethiopian National Health Care Quality and Safety Strategy (NQSS) (2021–2025) is to continually ensure and improve patient safety [17,18]. In September 2019, Ethiopia marks the first annual World Patient Safety Day by adding its voice to the international call, "Patient Safety: a global health priority" [7]. This event occurred approximately 15 years after the World Health Organization established the World Alliance for Patient safety in 2004 [7,19]. In Ethiopia, patient safety culture is a relatively new focus, and the implementation gap remains a long-standing concern [17,20,21]. The Ethiopian Institute for Healthcare Improvement has trained over 3,700 healthcare providers to improve safety standards, yet challenges including poor communication, insufficient training, poor reporting systems and lack of coordination persist [22].

Recognizing the frequency and nature of errors in primary care is crucial for formulating policy to reduce harm, enhancing patient safety, and improving the overall quality of healthcare. Patient safety culture is essential when it comes to the competencies of nursing and midwifery. Although there is an increasing focus on patient safety policies in low-income countries, there remains a paucity of research examines the view of healthcare practitioners in these settings [9,15,23]. Nationally representative data is needed to develop and implement a sustained culture among health professional, including nursing and midwifery care providers, to ensure patient safety as a priority. Therefore, the present study is designed to consolidate available data to determine the patient safety culture among health care providers in Ethiopia. As a result, the current systematic review and meta-analysis can be used as an important step in obtaining a clear view of patient safety aspects requiring urgent attention and developing national strategies to enhance the quality of care provided to patients. Moreover, understanding the current Ethiopian patient safety culture may present an opportunity to understand patient safety related issues across a wider Ethiopian audience, especially at the point of care.

## Materials and methods

### Study design and reporting

A systematic review and meta-analysis of observational studies conducted using a standardized tool, the Hospital Survey on Patient Safety Culture (HSOPSC), was conducted on patient safety culture among health care providers. All studies on patient safety culture, using HSOPSC tool, among health care providers in Ethiopia published up to October 10, 2024 (search updated on March 29, 2025) were retrieved using the Preferred Reporting Items for the Systematic Reviews and Meta-Analyses (PRISMA) updated guidelines (Supplementary file 1) [24].

### Protocol and registration

The protocol has been registered with PROSPERO (International Prospective register for Systematic Review), the University of York Center for Reviews and Dissemination with registration number CRD42023407601.

### Search strategy

We conducted a systematic and comprehensive search in the electronic databases of ScienceDirect, Medline, African Journals Online (AJOL), Excerpta Medical Database (EMBASE), Scopus and Google scholar to identify all the relevant observational studies on patient safety culture among health care providers in Ethiopia up to Oct 10, 2024. We have updated the search on the 29th of March 2025. To find additional potentially applicable studies, a manual search was conducted using reference lists of retrieved articles, and when required we contacted the authors. The primary author reached out via email to corresponding authors regarding incomplete data. Studies were confined to English language

publications. Search terms used include: "Patient", "Safety", "Culture", "Patient safety", "Safety Culture", "Patient safety culture", "Medical Error", "Adverse Events", "Health care providers", "Health workers", and "Ethiopia". Studies that assessed the prevalence of patient safety culture were considered relevant.

Our search limits restricted studies to those that were observational in nature, published in the English language, and involving human subjects. The PubMed search engine with MeSH (Medical Subject Headings) and Boolean operators were used to search the Medline database and presented as follows: (((((((((((((knowledge[MeSH Terms]) OR (Attitude[MeSH Terms])) OR (Prevalence[MeSH Terms])) AND (Patient safety[MeSH Terms])) AND (Safety culture[MeSH Terms])) OR (Patient safety culture[MeSH Terms])) AND (Healthcare providers[MeSH Terms])) OR (Midwives[MeSH Terms])) OR (Nurse[MeSH Terms])) AND (Government hospital[MeSH Terms])) OR (Health facility[MeSH Terms])) AND (Hospital Survey[MeSH Terms])) AND (Ethiopia)

### Eligibility criteria

Studies were included if they fulfilled the following criteria: i) Study period: studies conducted or published until Oct 10, 2024 (updated on March 29, 2025); ii) Study type: observational studies (cross-sectional, case control and cohort studies) using HSOPSC tool; iii)Participants (P): studies with a study population comprising health care providers; iv) Intervention (I): not applicable; v) Comparison (C): not applicable; vi) Outcome (O): studies that measured patient safety culture; vii) Place of study: studies conducted in Ethiopia; and vi) Studies published in the English language. Review articles, case series, case reports, and letters to editors were excluded.

### Study selection and extraction

Retrieved studies were imported into Endnote (Version X7, for Windows, Thomson Reuters, Philadelphia, PA, USA) and duplicated studies were removed. Two independent reviewers (GAA and BTN) screened all the papers for eligibility criteria: first, abstract and title and second, full text screening. If consensus could not be reached regarding study selection, disagreements were resolved by inviting a third investigator (YSA). Data was extracted by the same two researchers (GAA and BTN) using a standardized data extraction format prepared in Microsoft Excel. Variables included: first author's name, year of publication, region, study design, data collection method, sample size, total number of cases, and prevalence. If the study did not present the total number of cases, we used the raw data, including sample size and prevalence to calculate the number of cases.

### Quality assessment

After removing duplicates and the full-text review, methodological quality was assessed for each included study using the Newcastle-Ottawa Quality Assessment Scale checklist (adapted for cross-sectional studies) [25], comprising patient selection, study group comparability and outcome assessment. Three authors (GAA, BTN and YSA) assessed quality of articles. The review procedure was repeated whenever disagreement happened, with discrepancies resolved through discussion. The following items from the tool were used as criteria for quality appraisal: 1) representativeness of the sample, 2) sample size adequacy, 3) non-respondents, 4) ascertainment of the exposure (risk factors), 5) the subjects in different outcome groups are comparable based on study design, 6) assessment of the outcome and 7) statistical test (Supplementary file 2). The score ranged from 0 (lowest degree) to 9 (highest degree), and articles that achieved a score of five or lower on the quality assessment checklist criteria were classified as having a poor quality, six to seven stars to be adequate to good quality, and of eight to nine to be excellent quality. Our quality assessment result indicated that two studies [17,26] score nine, six studies [2,18,20,21,27,28] score eight, four studies [29–32] score seven and two studies [33,34] score six. The overall analysis of the score distribution in the quality assessment reveals that most studies were rated as high quality, while only a limited number of studies exhibited lower scores. To assess how low-quality studies impacted the result, subgroup analysis were conducted, stratified by NOS quality rating.

## Statistical analysis

The data were analyzed using STATA 15 software (StataCorp, College Station, Texas, USA). $I^2$ statistical test was computed to check heterogeneity across studies. $I^2$ values of 0%, 25%, 50%, and 75% were assumed to represent no, low, medium, and high heterogeneity, respectively. Since significant heterogeneity was detected between studies ($p < 0.001$, $I^2 = 91.3\%$), a meta-analysis using a random effects model was conducted to estimate pooled prevalence with 95% confidence intervals (CI). For articles reporting no magnitude for patient safety culture, we calculated them if sufficient information was provided. To determine the pooled odds ratio with 95% CI, we used a weighted inverse variance random effects model because of observed evidence of between study heterogeneity. For all tests, *p*-values are two-tailed and a *p*-value of 0.05 was used to determine the significance of the small study effect.

## Results

### Search results

A total of 667 articles were retrieved using the electronic databases. Fifty seven articles were deleted due to duplication. Of the remaining 610 articles, 472 were removed by title and abstract, while 138 were read in full and assessed for eligibility. Of the 124 studies, 96 were conducted outside of Ethiopia, 25 were conducted on different population, and 3 studies did not provide data amenable to meta-analysis. Finally, 14 studies with a total of 5,901 healthcare providers met

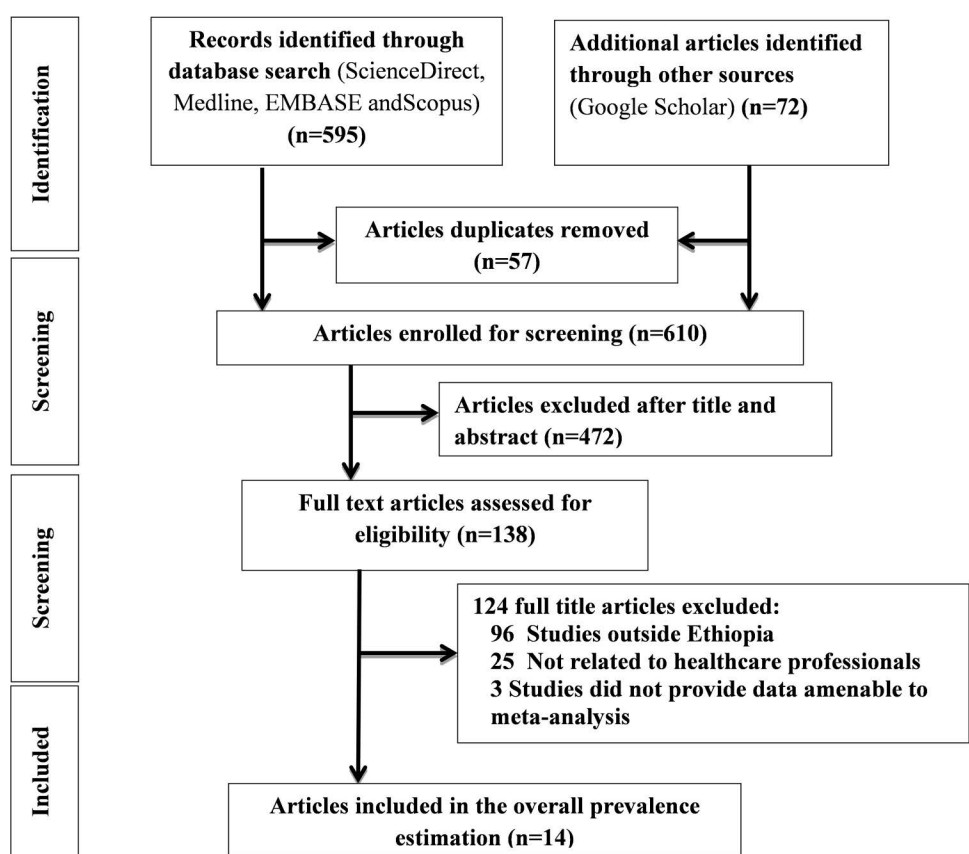

**Fig 1. PRISMA flow diagram of articles screened and the selection process on patient safety culture among health care providers in Ethiopia.**

the eligibility criteria and were entered into the meta-analysis process. Steps of the search strategy are shown in Fig 1. Descriptions of 610 excluded records and reasons are presented in S3 file (Supplementary file 3).

## Characteristics of included studies

Among the 14 included studies, six were from Amhara [2,17,21,27,29,30], four were from Oromia [20,26,28,33], two from Addis Ababa [18,34], and one each from Southern Nations, Nationalities and Peoples' Region [31], and Tigray [32]. The study from Amhara [30] documented the highest prevalence (58.4%) of patient safety culture among health care providers, while the lowest prevalence (21.6%) was observed from a study conducted in the Tigray region [32]. Table 1 summarizes the characteristics of the included studies.

**Table 1. Summary of studies included in meta- analysis that shows the prevalence of patient safety culture among health care professionals in Ethiopia, 2024.**

| Authors | Public-ation year | Date of data extraction | Name of data extractor/s | Region | Study design | Data collec-tion method | Sample size (n) | RR[a] (%) | Cases | Prevale-nce | Eligibility status |
|---|---|---|---|---|---|---|---|---|---|---|---|
| Mohamed, F. et al [2] | 2021 | Oct 14, 2024 | Negash BT | Amhara | CS[b] | SA[c] | 411 | 97.4 | 184 | 44.8 | Eligible |
| Wami, SD. et al [26] | 2016 | Oct 14, 2024 | Negash BT | Oromia | CS | SA | 596 | 93.6 | 278 | 46.7 | Eligible |
| Kumbi, M. et al [28] | 2020 | Oct 14, 2024 | Asgedom YS | Oromia | CS | SA | 518 | 93.2 | 228 | 44.0 | Eligible |
| Ayisa, A. et al [29] | 2021 | Oct 14, 2024 | Azeze GA | Amhara | CS | Interview | 530 | 92.2 | 240 | 45.3 | Eligible |
| Mekonnen, Et al [21] | 2018 | Oct 14, 2024 | Azeze GA | Amhara | CS | SA | 410 | 85.4 | 189 | 46.0 | Eligible |
| Garuma, M. et al [20] | 2015 | Oct 14, 2024 | Azeze GA | Oromia | CS | SA | 388 | 92.2 | 191 | 49.2 | Eligible |
| Mitiku M et al [32] | 2015 | Oct 14, 2024 | Asgedom YS | Tigray | CS | SA | 323 | 99.4 | 70 | 21.6 | Eligible |
| Yismaw W et al [33] | 2023 | Oct 14, 2024 | Negash BT | Oromia | Mixed method | SA | 395 | 89.98 | 158 | 40.0 | Eligible |
| Ayanaw T et al [30] | 2023 | Oct 14, 2024 | Negash BT | Amhara | CCS[d] | SA | 310 | 99.4 | 181 | 58.4 | Eligible |
| Shashamo et al [31] | 2023 | Oct 14, 2024 | Negash BT | SNNPR[e] | CS | SA | 398 | 97.55 | 202 | 50.8 | Eligible |
| Yayehrad T et al [18] | 2024 | Oct 14, 2024 | Negash BT | Addis Ababa | CS | SA | 461 | 93.3 | 228 | 49.5 | Eligible |
| Ali M et al [27] | 2024 | Feb 7, 2025 | Azeze GA | Amhara | CS | SA | 422 | 100 | 211 | 50.1 | Eligible |
| Mulugeta TT [34] | 2019 | Feb 7, 2025 | Azeze GA | Addis Ababa | CS | SA | 346 | 74.7 | 152 | 44 | Eligible |
| Atinafu D et al [17] | 2024 | Feb 7, 2025 | Azeze GA | Amhara | CS | SA | 393 | 93.6 | 187 | 47.6 | Eligible |

[a]RR: Response Rate,

[b]CS: Cross-sectional,

[c]SA: Self-administered,

[d]CCS: Comparative Cross-sectional,

[e]SNNPR: Southern Nations, Nationalities and People's Region.

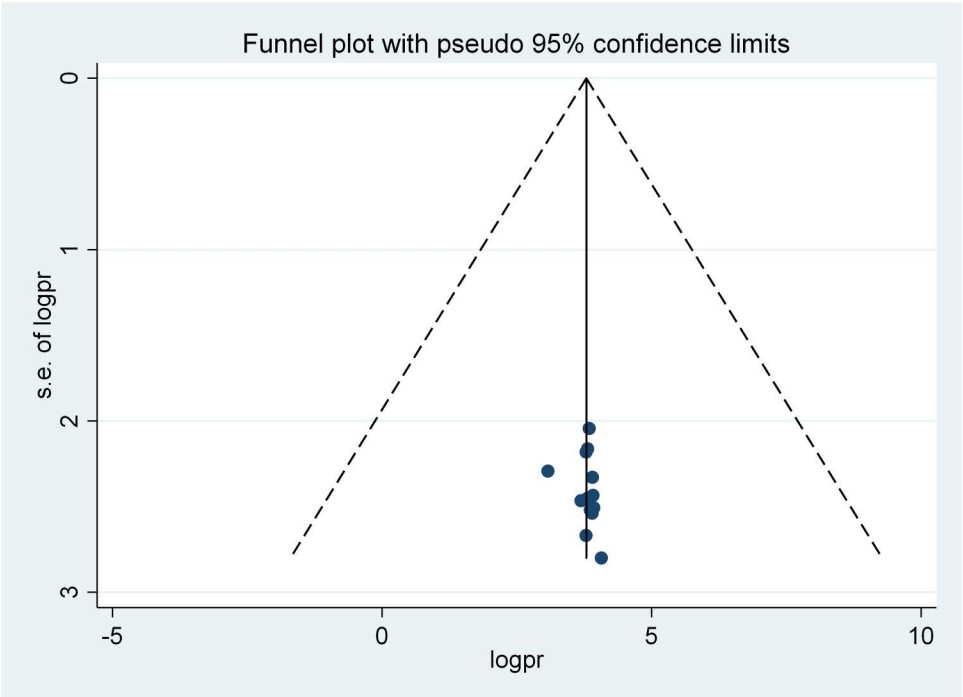

**Fig 2. Funnel plot of publication bias using data from 14 studies.** Each dot represents a study. ES = effect size, s.e = standard error.

## Heterogeneity and publication bias

Significant heterogeneity was observed among the studies in terms of the pooled prevalence of patient safety culture ($I^2$ = 91.3%, p-value<0.001). To address this, meta-analysis using random effects model was computed to estimate the pooled prevalence. Graphical distribution of the Funnel plot shows evidence of asymmetry (Fig 2). Moreover, finding from Egger's test revealed that the estimated bias coefficient (intercept) is 0.291 with standard error of 0.313, giving a *p* value of 0.370 at 95% confidence interval. The test thus provides strong evidence for the absence of publication bias.

## Patient safety culture among health care providers in public hospitals of Ethiopia

In the random effects meta-analysis, the pooled prevalence of patient safety culture among Health care providers in Ethiopia was 45.52% (95% CI: 41.27, 49.76; $I^2$ = 91.3; p-value<0.001). Fig 3 presents forest plot of the 14 included studies.

## Subgroup analysis

Significant heterogeneity ($I^2$ = 91.3%; p<0.001) was noted among the included studies, prompting a subgroup analysis to compare the prevalence estimates of patient safety culture across different groups. We stratified by predefined subgroups namely geographic region, year of publication, sample size and Newcastle-Ottawa Scale quality rating; those rated as excellent (greater than or equal to eight) [2,17,18,20,21,26–28] and those deemed to have adequate quality (fewer than eight stars) [29–34]. Subgroup analysis indicated the highest weighted prevalence of patient safety culture was reported from Amhara region (n = 6, 48.17%, 95% CI 46.20, 50.13, *p* = 0.003), those studies published after 2019 (n = 10, 47.54, 95% CI 46.04, 49.04, *p*<0.001), in the larger NOS ratings (n = 8, 47.61, 95% CI 45.86, 49, 36, *p* = 0.003). In the nine studies with lower than 420 study participants, patient safety culture was lower (n = 9, 43.89, 95% CI 42.25, 45.53, *p*<0.001) (Table 2).

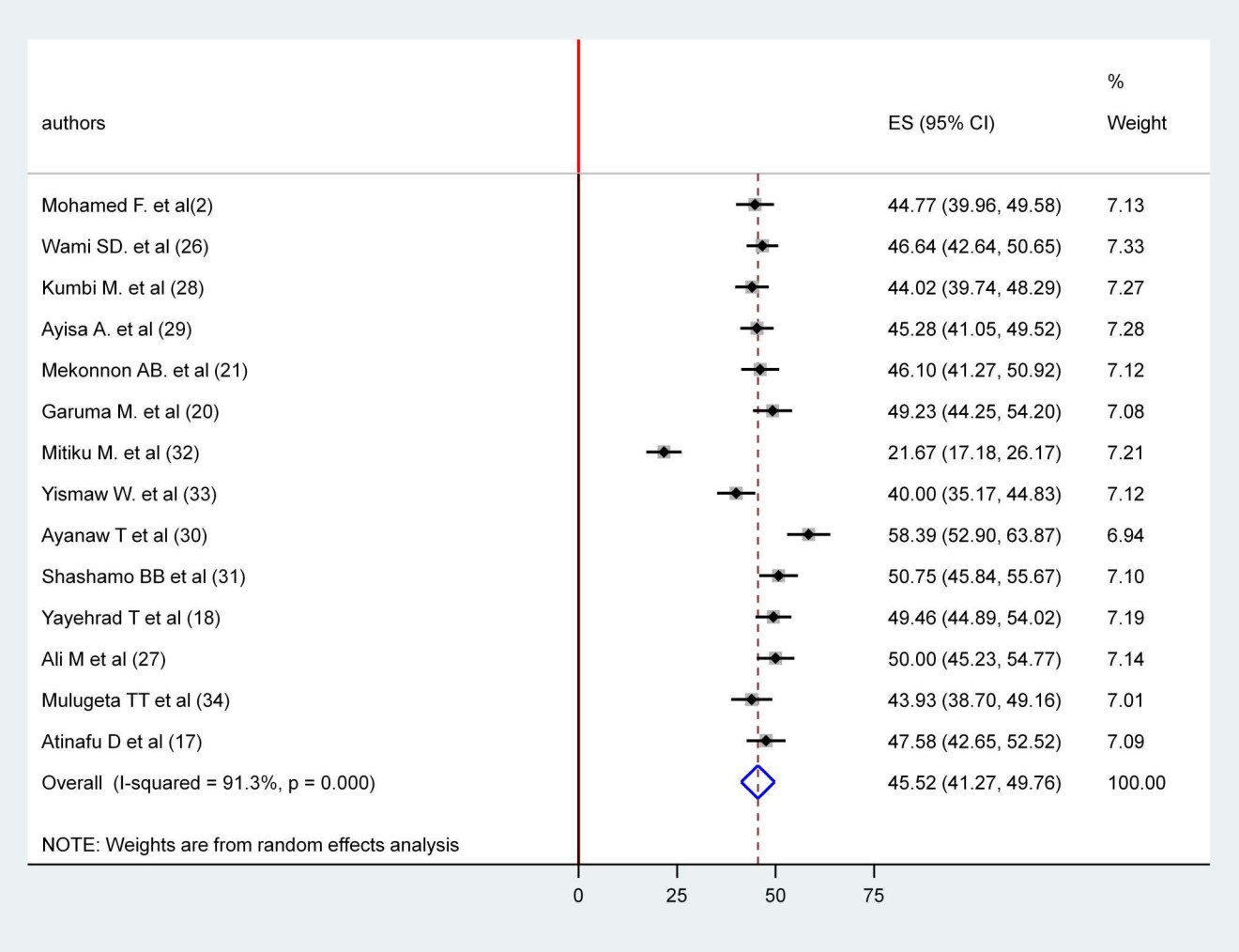

**Fig 3. Forest plot of patient safety culture among health care providers in public hospitals of Ethiopia, 2024.**

## A leave-out-one sensitivity analysis

To ensure that the pooled estimates of patient safety culture results were not affected by one or more several studies, leave-out-one sensitivity analysis method was used that involved performing the analysis on the data by leaving out one study at a time. The results of sensitivity analysis, using random effects model, suggested that the excluded study did not bring any significant change to the overall prevalence estimate of patient safety culture practice in public hospitals of Ethiopia (Fig 4).

## Discussion

There is a clear consensus that quality health services across the world should be safe, effective and centered around the needs of individuals [11]. Ensuring patient safety is fundamental component in the provision of high-quality essential health services. It aims to mitigate and minimize the risks, errors and harm that occur to patients during provision of healthcare. In Ethiopia, the concept of patient safety culture is a relatively new focus and little is known about the current

**Table 2. Subgroup analysis on patient safety culture among healthcare providers in Ethiopia.**

| Subgroups | Category | Number of studies | Effect size (95% CI) | I-square | *P*-value |
|---|---|---|---|---|---|
| **Region** | Amhara | 6 | 48.17 (46.20, 50.13) | 72.8% | 0.003 |
| | Oromia | 4 | 45.03 (42.79, 47.26) | 61.2% | 0.052 |
| | Addis Ababa | 2 | 47.07 (43.63, 50.51) | 58.9% | 0.119 |
| | Tigray | | | | |
| | SNNPR[b] | | | | |
| **Publication Year** | ≤2019 | 4 | 39.54 (37.25, 41.83) | 96.4% | <0.001 |
| | >2019 | 10 | 47.54 (46.04, 49.04) | 72.8% | <0.001 |
| **Sample size** | <420 | 9 | 43.89 (42.25, 45.53) | 94.2% | <0.001 |
| | ≥420 | 5 | 46.88 (44.94, 48.83) | 22.4% | 0.272 |
| **NOS[a] rating** | <8 | 7 | 42,54 (40.74, 44.33) | 94.7% | <0.001 |
| | ≥8 | 7 | 47.61 (45.86, 49, 36) | 70.2% | 0.003 |

[a]**NOS Newcastle-Ottawa Scale**,

[b]**Southern Nations, Nationalities and People's Region.**

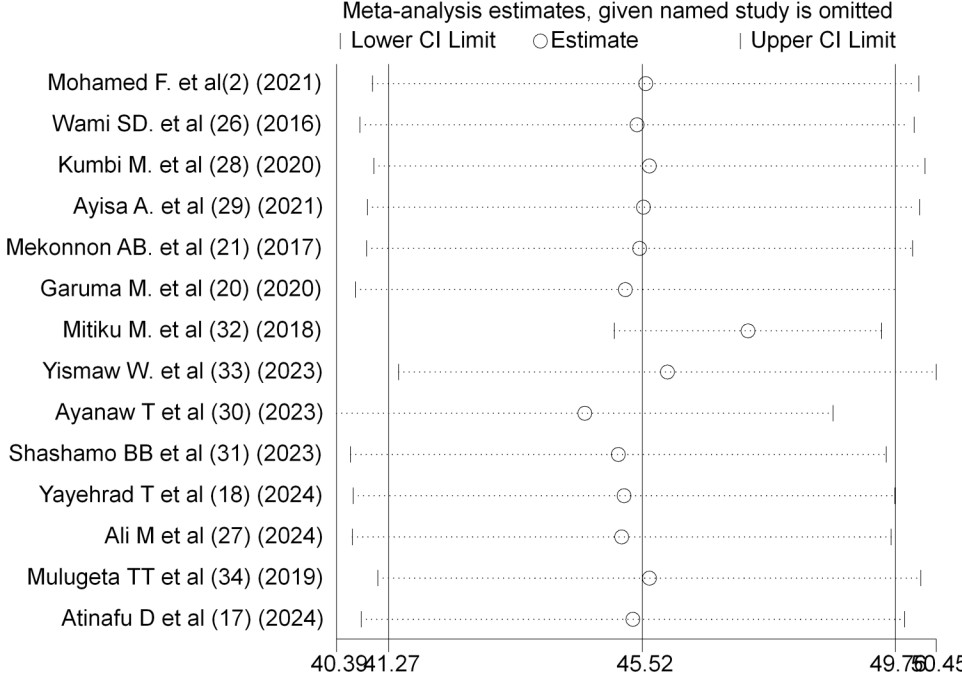

**Fig 4. Result of sensitivity analysis of the 14 included studies.**

status of public hospitals patient safety culture in the country. Hence, we conducted this systematic review and meta-analysis to estimate the pooled prevalence of patient safety culture among healthcare providers in Ethiopia.

The current study demonstrated a significant level of statistical heterogeneity ($I^2 = 91.3\%$; $p < 0.001$), which impacts the reliability of the overall estimates. Several factors maycontribute may contribute to this variability, including the participation of various groups of healthcare professionals, small studies, differences in geographic origin and unmeasured

characteristics. The notable heterogeneity identified in our study highlights the importance of not just the overall pooled estimates, but also the individual estimates from each study.

A total of 14 studies, selected based on the series of inclusion criteria, were included into the analysis. Based on the meta-analysis result, the pooled prevalence of patient safety culture among health care providers in Ethiopia was 45 percent. The causes for the observed poor patient safety in Ethiopia may be attributable to different factors, including lack of effective organizational leadership, lack of resources, poor learning resources, insufficient availability and implementation of protocols and policies, communication and reporting gaps. Our finding provides insight to pay the closest possible attention to the problem of patient safety and to have strategies and organizational structure to deliver coordinated care to ensure that the care received by patients is of high quality and effective, as well as safe.

This study finding is nearly comparable with the finding from a systematic review and meta-analysis conducted in Iran [35] and Latin American hospitals [36], where the overall pooled prevalence of patient safety culture was 50.5 percent and 48 percent, respectively. The current finding is also similar with a study conducted in South Africa, where the overall prevalence of patient safety culture was 42.4% [37]. Understandably, concerns related to poor staffing may negatively impact patient safety as has been reported from studies in Ethiopia and Latin American countries [34,36]. Moreover, the observed poor patient safety culture in Ethiopia, like that of many other developing nations in the world, might be influenced by institutional factors including lack of commitment by hospital managers and administrators, shortage of financial resources and supplies, lack of information, shortage of medical expertise and poor staffing [19].

Our study result is lower than a study conducted using the same standardized tool, the Hospital Survey on Patient Safety Culture (HSOPSC), among health workers representing five secondary and tertiary care hospitals in Oman, where the overall prevalence of patient safety culture was 58% [38](39). In addition, our finding is also lower than other studies conducted in Sri Lanka [39] and 42 hospitals in Taiwan [40], where the overall prevalence of patient safety culture was 81.3 and 64 percent, respectively. This variation might be due to the difference in the context of health care provision, infrastructure, hospital leadership and management and resourcing across countries. Taiwan, for instance, is a high income country; therefore it is not surprising to see a strong safety culture in a variety of high-income countries. The same conclusion was made by Rice HE. et al [41], where it was found that different programs to modify the safety culture have led to lasting improvements in patient safety and quality of care in high income settings around the world.

A study by Brian Yu et al [42], assessed national trends in PSC among Taiwanese hospital staff, revealing that dimensions of patient safety culture significantly increased over a period of eight years. Similarly, another study conducted in Taiwan [43], which was aimed to investigate the role of management leadership in promoting a culture of patient safety revealed that effective management leadership plays a pivotal role in shaping safety attitudes and improve overall quality of care.

### Implication for research, policy and practice

The current sub-optimal patient safety culture has serious implications

**Implications for medical education.** To enhance the knowledge, skills and behaviors of HCPs in Ethiopia, Medical education programs should incorporate more patient safety content into curriculums. Development of supportive faculty and continuing education actions is essential. Indeed, the WHO has developed the Patient Safety Curriculum guide for medical schools that is designed to be either integrated into existing undergraduate/graduate medical curricula or offered as independent modules, regardless of geographic or cultural contexts [19,44].

**Implications for hospital managers and healthcare administrators.** Hospital managers and healthcare leaders should stable recruitment programs and maintain regular communication among staff regarding patient safety. Quality leaders should prioritize training initiatives by making patient safety discussions a top agenda at seminars and conferences. Moreover, hospital management should implement continuous quality improvement strategies, focusing on

prioritizing goals while optimizing limited resources. They should create reward systems for reporting unsafe practices and promoting patient safety, ensuring the availability and functionality of equipment and supplies, and establish minimum staffing standards.

**Implications for strategies and policymakers.** Should implement quality improvement interventions to enhance the quality and safety of care, which can positively influence PSC. Policy makers should establish a comprhensive regulatory framework that encompasses safe clinical systems, processes, policies, evidence-based practices, regulations with strong enforcement mechanisms.

**Implications for HCPs.** HCPs should focus on the following activities: engagement from all the staff levels, enhancing interpersonal communication, pursuing ongoing professional development, delivering timely services, participating in teamwork education programs and fostering a culture of accountability within their organizations.

**Implication for researchers.** To better understand patient safety culture, a survey to highlight the overall areas of concern supplemented with interviews with the staffs to provide a description of what the problem involved is important to understand the details and to be able to choose the right solution.

### Strength and limitations of the study

**Strengths of the study.** The strength of our study lies in the fact that this was, to our knowledge, the first systematic review and meta-analysis to determine the overall prevalence of patient safety culture among health care providers and all the studies included were of good methodological quality. To minimize the risks of publication bias, comprehensive literature searches were conducted. Further, this systematic review and meta-analysis was performed and reported according to PRISMA guideline.

**Limitation of the study.** $I^2$ test shows a significant heterogeneity. Therefore, we applied random-effect analysis model.. The high heterogeneity reported in the current study limits confidence in the robustness of conclusions. We acknowledge the potential social desirability bias resulting from the HSOPSC measurement tools utilized in the primary studies. One of the other significant limitations of our study is the potential for publication bias, as it only includes observational studies published in English language. Our finding may also be subject to unpublished studies. The current systematic review and meta-analysis has not assessed for factors associated with patient safety culture.

## Conclusion

The reported patient safety culture among health care providers in Ethiopia remained poor. Therefore, based on the evidence, this review recommends more evidence-based proactive project planning and implementation focusing on education, training and development of guidelines to integrate a patient safety culture into the existing health system. Health services should aim to improve patient safety practiced by various healthcare professionals. This can be accomplished by integrating patient safety education and training topics early in pre-service curricula.

## Supporting information

**Supplementary file 1. PRISMA 2020 Checklist.**
(DOCX)

**Supplementary file 2. Methodological quality assessment of 14 included studies using the Newcastle-Ottawa quality assessment scale.**
(DOCX)

**Supplementary file 3. Description of 610 excluded records and reasons for exclusion.**
(XLSX)

## Acknowledgments

We acknowledge the authors of primary studies included in this review.

## Author contributions

**Conceptualization:** Gedion Asnake Azeze.

**Data curation:** Gedion Asnake Azeze.

**Formal analysis:** Gedion Asnake Azeze, Kirubel Eshetu Haile, Amanuel Yosef Gebrekidan, Gizachew Ambaw Kassie.

**Methodology:** Gedion Asnake Azeze, Berhan Tsegaye Negash, Yordanos Sisay Asgedom.

**Project administration:** Gedion Asnake Azeze, Berhan Tsegaye Negash, Yordanos Sisay Asgedom.

**Software:** Gedion Asnake Azeze, Kirubel Eshetu Haile, Amanuel Yosef Gebrekidan, Gizachew Ambaw Kassie, Berhan Tsegaye Negash, Yordanos Sisay Asgedom.

**Supervision:** Gedion Asnake Azeze, Kirubel Eshetu Haile, Amanuel Yosef Gebrekidan, Gizachew Ambaw Kassie.

**Validation:** Gedion Asnake Azeze, Kirubel Eshetu Haile, Amanuel Yosef Gebrekidan, Gizachew Ambaw Kassie.

**Visualization:** Gedion Asnake Azeze, Kirubel Eshetu Haile, Amanuel Yosef Gebrekidan, Gizachew Ambaw Kassie, Berhan Tsegaye Negash, Yordanos Sisay Asgedom.

**Writing – original draft:** Gedion Asnake Azeze, Kirubel Eshetu Haile, Amanuel Yosef Gebrekidan, Gizachew Ambaw Kassie, Berhan Tsegaye Negash, Yordanos Sisay Asgedom.

**Writing – review & editing:** Gedion Asnake Azeze, Kirubel Eshetu Haile, Gizachew Ambaw Kassie, Berhan Tsegaye Negash, Yordanos Sisay Asgedom.

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
