## [Decision Letter · Decision Letter 0]

9 Jan 2025

PONE-D-24-47092Patient safety in public hospitals of Ethiopia: a systematic review and meta-analysisPLOS ONE

Dear Dr. Haile,

Thank you for submitting your manuscript to PLOS ONE. After careful consideration, we feel that it has merit but does not fully meet PLOS ONE’s publication criteria as it currently stands. Therefore, we invite you to submit a revised version of the manuscript that addresses the points raised during the review process.

We look forward to receiving your revised manuscript.

Kind regards,

Maher Abdelraheim Titi

Academic Editor

PLOS ONE

Journal Requirements:

Reviewers' comments:

Reviewer's Responses to Questions

**Comments to the Author**

1. Is the manuscript technically sound, and do the data support the conclusions?

Reviewer #1: Yes

Reviewer #2: No

Reviewer #3: Yes

2. Has the statistical analysis been performed appropriately and rigorously? 

Reviewer #1: Yes

Reviewer #2: Yes

Reviewer #3: Yes

3. Have the authors made all data underlying the findings in their manuscript fully available?

Reviewer #1: Yes

Reviewer #2: No

Reviewer #3: Yes

4. Is the manuscript presented in an intelligible fashion and written in standard English?

Reviewer #1: Yes

Reviewer #2: No

Reviewer #3: Yes

5. Review Comments to the Author

Reviewer #1: The author has followed the standard procedures in preparation of the manuscript. The study is scientifically sound and reliable. The methodology was sound ensuring validity of the results and the presentation of the findings and discussion is appropriate. Patient safety culture is a topical issue that needs to be explored and efforts should be made to improve it in order to achieve universal health coverage. The findings from the study should stimulate a deeper dive into the factors associated with sub optimal patient safety culture in Ethiopia and the African region.

Reviewer #2: • Clarity of Objectives:

They lack clear delineation, making it less accessible to readers.

• Abstract:

The abstract is comprehensive but dense. Key details such as the number of studies, methodology, and major findings could be presented more concisely.

• Consistency:

Phrases like "patient safety culture" and "culture of patient safety" are used interchangeably, potentially confusing readers.

• Background:

The introduction provides extensive global context but offers limited background on the Ethiopian healthcare system's specific challenges related to patient safety.

• Quality Assessment:

While the manuscript mentions the use of the Newcastle-Ottawa Scale, it does not detail the quality ratings or how low-quality studies impacted the results.

• Heterogeneity and Publication Bias:

The study reports high heterogeneity (I² = 93.9%), which is expected given the variability in healthcare settings across Ethiopia. However, the implications of such heterogeneity on the reliability of pooled estimates are not discussed.

• Implications and Recommendations:

Recommendations are general (education, training, guidelines) and lack specificity regarding implementation within Ethiopia’s resource constraints.

• Language and Grammar:

Grammatical errors and awkward phrasing detract from the manuscript's professionalism. For example:

o "Due to unsafe care, 134 million adverse events occur in hospitals in LMICs each year resulting in 2.6 million deaths" could be rephrased for clarity. Conduct a thorough language review or use professional editing services to polish the manuscript.

• Results

The narrative could benefit from restructuring for clarity. For example:

o Group statistical findings (e.g., prevalence, heterogeneity) separately from qualitative interpretations.

o Add a clear flow to the explanation of subgroup and sensitivity analyses.

• Addressing Heterogeneity:

Significant heterogeneity (I² = 93.0%) remains a limitation, and while the random-effects model mitigates this, further exploration into the sources of heterogeneity would strengthen the paper. Were differences in survey methodology, sample size, or health system infrastructure contributors?

• Discussion:

While the comparison to other countries is helpful, the discussion could elaborate on the root causes of Ethiopia's low patient safety culture prevalence (e.g., resource constraints, health system gaps, or cultural attitudes toward patient safety). The implications for practice should specify who will lead the recommended changes (e.g., government agencies, hospital administrators) and what key steps are needed for implementation. Ensure that all references are cited in alignment with journal requirements. Some critical comparisons (e.g., studies from Oman, Taiwan, and South Africa) could include more detailed descriptions of how those countries addressed patient safety culture gaps.

• Limitations Section:

The discussion of limitations is brief. Consider addressing potential publication bias (despite Egger’s test) and the challenges of using self-reported measures like the HSOPSC, which may introduce social desirability bias.

Reviewer #3: The author writes the article in a rigorous and standard language fashion.

the author selected and extracted based on the protocol and those who have used tools for data collection.

Major comments:

the author wrote different data collection periods: line 26 vs. line 104

The author report six articles are screened based on incomplete data but don't report whether contacted for obtaining and confirming data from investigators. Do you exclude them with a criteria?

The authors write the title in two phrases, alternatively "patient safety' and 'patient safety culture'. Which one is correct to align the outcome?

Did the author check these articles?

Ali M, Ademe S, Shumiye M, Hamza A. Patient safety culture and associated factors among health care workers in south Wollo zone public hospitals, north east Ethiopia. Perioperative Care and Operating Room Management. 2024;35:100374. doi:10.1016/j.pcorm.2024.100374

Mulugeta, Tigist Tedla. “Patient Safety Culture among Health Workers in Addis Ababa Regional Hospitals, Ethiopia.” TEXILA INTERNATIONAL JOURNAL OF PUBLIC HEALTH, vol. 7, no. 2, June 2019, pp. 153–66, https://doi.org/10.21522/tijph.2013.07.02.art017.

6. PLOS authors have the option to publish the peer review history of their article (what does this mean? ). If published, this will include your full peer review and any attached files.

**Do you want your identity to be public for this peer review?** For information about this choice, including consent withdrawal, please see our Privacy Policy .

Reviewer #1: **Yes: ** Dr Abduljalil Umar Abdullahi

Reviewer #2: No

Reviewer #3: No

---

## [Author Response · Author response to Decision Letter 0]

1 Apr 2025

Date: Feb 17, 2025

Dr Maher Abdelraheim Titi

Academic Editor

PLOS ONE

RE: PONE-D-24-47092 Patient safety in public hospitals of Ethiopia: a systematic review and meta-analysis

Dear Dr Maher,

Thank you for considering our manuscript and for arranging for it to be reviewed by reviewer. We have tried to address your comments and the comments / suggestions from the reviewers.

Please find for your kind consideration the following:

 In the Response to Reviewers, we copy each of the comments / suggestions and provide the RESPONSE underneath (below pages 2-6) - uploaded as “Response to reviewers”.

 We also provide a marked-up copy of the manuscript that highlights changes made to the original version and this is uploaded as a separate file labelled “Revised Manuscript with Track Changes”.

 In our point-by-point response, we referred to the “Revised Manuscript with Track Changes” document for page and line numbers.

We have been carefully through the peer review and have revised our paper accordingly. We feel that the paper is much improved as a result of this peer review process, and thank you for taking it to this stage.

While hoping that these changes would meet with your favourable consideration, we hold ourselves at your entire disposition for any further information or other changes you might require.

Best wishes

Kirubel Eshetu Haile; on behalf of the co-authors

POINT BY POINT RESPONSE TO THE EDITOR AND REVIEWERS SUGGESTIONS

Journal Requirements:

RESPONSE: manuscript preparation follows PLOS ONE's style requirements and file naming done as requested.

2. PLOS requires an ORCID ID for the corresponding author in Editorial Manager on papers submitted after December 6th, 2016. Please ensure that you have an ORCID ID and that it is validated in Editorial Manager. To do this, go to ‘Update my Information’ (in the upper left-hand corner of the main menu), and click on the Fetch/Validate link next to the ORCID field. This will take you to the ORCID site and allow you to create a new ID or authenticate a pre-existing ID in Editorial Manager.

RESPONSE: Done as requested.

Reviewer Comments:

We thank the reviewers for reviewing this paper and for their comments and suggestions.

The different points raised by the reviewers include:

Response to Reviewer #1

The author has followed the standard procedures in preparation of the manuscript. The study is scientifically sound and reliable. The methodology was sound ensuring validity of the results and the presentation of the findings and discussion is appropriate. Patient safety culture is a topical issue that needs to be explored and efforts should be made to improve it in order to achieve universal health coverage. The findings from the study should stimulate a deeper dive into the factors associated with sub optimal patient safety culture in Ethiopia and the African region.

RESPONSE: Thank you for your comment. We believe that the manuscript has significantly benefited from the peer review process.

Response to Reviewer #2

1. Clarity of Objectives: They lack clear delineation, making it less accessible to readers.

RESPONSE: We made changes and we have added the aim of the study in the last statement of background part of the abstract section.

2. The abstract is comprehensive but dense. Key details such as the number of studies, methodology, and major findings could be presented more concisely.

RESPONSE: All of the mentioned items were taken into consideration and changes have been made as requested.

3. Consistency: Phrases like "patient safety culture" and "culture of patient safety" are used interchangeably, potentially confusing readers.

RESPONSE: We have made changes accordingly and we tried to be consistent throughout the document.

4. Background: The introduction provides extensive global context but offers limited background on the Ethiopian healthcare system's specific challenges related to patient safety.

RESPONSE: Thank you for highlighting this. A paragraph has been added, particularly stating the Ethiopian context (page number 5; line number 89-98).

5. Quality Assessment: While the manuscript mentions the use of the Newcastle-Ottawa Scale, it does not detail the quality ratings or how low-quality studies impacted the results.

RESPONSE: Changes have been made as requested. Additionally, we have conducted subgroup analysis based on the results of Newcastle-Ottawa Scale rating (Page 9; line number 180-188).

6. Heterogeneity and Publication Bias: The study reports high heterogeneity (I² = 93.9%), which is expected given the variability in healthcare settings across Ethiopia. However, the implications of such heterogeneity on the reliability of pooled estimates are not discussed.

Response: We have added a paragraph describing the source of heterogeneity, statistical analysis and possible implication for the pooled estimates (Page 14; line number 265-270).

7. Implications and Recommendations:

Recommendations are general (education, training, guidelines) and lack specificity regarding implementation within Ethiopia’s resource constraints.

Response: In response to your comment (and your comments on discussion section) we added statements in discussion and implication sections; focusing recommendations on specific bodies (Page Number 15-17; line numbers 303-346) (Page Number 15-16; line numbers 303-308).

Language and Grammar:

Grammatical errors and awkward phrasing detract from the manuscript's professionalism. For example:

o "Due to unsafe care, 134 million adverse events occur in hospitals in LMICs each year resulting in 2.6 million deaths" could be rephrased for clarity. Conduct a thorough language review or use professional editing services to polish the manuscript.

Response: Done as requested.

8. Results

The narrative could benefit from restructuring for clarity. For example:

a. Group statistical findings (e.g., prevalence, heterogeneity) separately from qualitative interpretations.

b. Add a clear flow to the explanation of subgroup and sensitivity analyses.

Response: Thank you. We have made all necessary changes subgroup and sensitivity analysis explained clearly (Page 12 to 13; line number 232-253).

9. Addressing Heterogeneity:

Significant heterogeneity (I² = 93.0%) remains a limitation, and while the random-effects model mitigates this, further exploration into the sources of heterogeneity would strengthen the paper. Were differences in survey methodology, sample size, or health system infrastructure contributors?

Response: Thank you very much for highlighting this. After this comment, we conducted a subgroup analysis stratified by predefined subgroups namely geographic region, year of publication, sample size and Newcastle-Ottawa Scale quality rating. Result presented in table form (Page 12 to 13; line number 232-246).

10. Discussion:

While the comparison to other countries is helpful, the discussion could elaborate on the root causes of Ethiopia's low patient safety culture prevalence (e.g., resource constraints, health system gaps, or cultural attitudes toward patient safety). The implications for practice should specify who will lead the recommended changes (e.g., government agencies, hospital administrators) and what key steps are needed for implementation. Ensure that all references are cited in alignment with journal requirements. Some critical comparisons (e.g., studies from Oman, Taiwan, and South Africa) could include more detailed descriptions of how those countries addressed patient safety culture gaps.

Response: Thank you for your comment. In addition to comparing findings, we included insights on the causes of poor patient safety in Ethiopia and compared our result with those from South Africa and Taiwan with its implication (Page 14; line number 273-276).

11. Limitations Section:

The discussion of limitations is brief. Consider addressing potential publication bias (despite Egger’s test) and the challenges of using self-reported measures like the HSOPSC, which may introduce social desirability bias.

Response: limitation section modified accordingly.

Response to Reviewer #3:

1. The author writes the article in a rigorous and standard language fashion.

Response: Thank you very much

2. The author selected and extracted based on the protocol and those who have used tools for data collection.

Response: Thank you very much

Major comments:

3. The author wrote different data collection periods: line 26 vs. line 104

Response: Thank you. Corrections made

4. The author report six articles are screened based on incomplete data but don't report whether contacted for obtaining and confirming data from investigators. Do you exclude them with a criteria?

Response: Thank you for highlighting this. During our updated search, the corresponding authors were contacted through email for further information, particularly to obtain information on measurement and operational definition of outcome variable. We reported this on our methods section (page 7; line number 134-135).

5. The authors write the title in two phrases, alternatively "patient safety' and 'patient safety culture'. Which one is correct to align the outcome?

Response: Corrections has been made

6. Did the author check these articles?

i. Ali M, Ademe S, Shumiye M, Hamza A. Patient safety culture and associated factors among health care workers in south Wollo zone public hospitals, north east Ethiopia. Perioperative Care and Operating Room Management. 2024;35:100374. doi:10.1016/j.pcorm.2024.100374

ii. Mulugeta, Tigist Tedla. “Patient Safety Culture among Health Workers in Addis Ababa Regional Hospitals, Ethiopia.” TEXILA INTERNATIONAL JOURNAL OF PUBLIC HEALTH, vol. 7, no. 2, June 2019, pp. 153–66, https://doi.org/10.21522/tijph.2013.07.02.art017.

Response: Thank you very much. We have revised our search, and with the addition of these two studies along with one new study, the total number of studies included in the final meta-analysis has risen from 11 to 14.

Thank you!

---

## [Decision Letter · Decision Letter 1]

18 May 2025

Patient safety culture in public hospitals of Ethiopia: a systematic review and meta-analysis

PONE-D-24-47092R1

Dear Dr. Haile,

We’re pleased to inform you that your manuscript has been judged scientifically suitable for publication and will be formally accepted for publication once it meets all outstanding technical requirements.

Kind regards,

Maher Abdelraheim Titi

Academic Editor

PLOS ONE

Reviewers' comments:

Reviewer's Responses to Questions

**Comments to the Author**

1. If the authors have adequately addressed your comments raised in a previous round of review and you feel that this manuscript is now acceptable for publication, you may indicate that here to bypass the “Comments to the Author” section, enter your conflict of interest statement in the “Confidential to Editor” section, and submit your "Accept" recommendation.

Reviewer #3: All comments have been addressed

2. Is the manuscript technically sound, and do the data support the conclusions?

Reviewer #3: Yes

3. Has the statistical analysis been performed appropriately and rigorously? 

Reviewer #3: I Don't Know

4. Have the authors made all data underlying the findings in their manuscript fully available?

Reviewer #3: Yes

5. Is the manuscript presented in an intelligible fashion and written in standard English?

Reviewer #3: Yes

6. Review Comments to the Author

Reviewer #3: The authors have adequately addressed my comments raised in a previous round. The manuscript is edited and written in standard English. This manuscript is now acceptable for publication.

7. PLOS authors have the option to publish the peer review history of their article (what does this mean? ). If published, this will include your full peer review and any attached files.

**Do you want your identity to be public for this peer review?** For information about this choice, including consent withdrawal, please see our Privacy Policy .

Reviewer #3: No

---

## [Editor Report · Acceptance letter]

PONE-D-24-47092R1

PLOS ONE

Dear Dr. Haile,

I'm pleased to inform you that your manuscript has been deemed suitable for publication in PLOS ONE. Congratulations! Your manuscript is now being handed over to our production team.

Kind regards,

on behalf of

Dr. Maher Abdelraheim Titi

Academic Editor

PLOS ONE